# Targeting Protein Aggregation in ALS

**DOI:** 10.3390/biom14101324

**Published:** 2024-10-18

**Authors:** Michele Perni, Benedetta Mannini

**Affiliations:** 1Baz-Therapeutics Inc., 810 Rittenhouse Square, Suite 412, Philadelphia, PA 19103, USA; 2Clinical Research Building, Perelman School of Medicine, University of Pennsylvania, 415 Curie Boulevard, Philadelphia, PA 19104, USA; 3Department of Experimental and Clinical Biomedical Sciences Mario Serio, University of Florence, Viale Morgagni 50, 50134 Florence, Italy; 4Centre for Misfolding Diseases, Yusuf Hamied Department of Chemistry, University of Cambridge, Cambridge CB2 1EW, UK

**Keywords:** neurodegeneration, amyotrophic lateral sclerosis, drug discovery, clinical trials, protein aggregation, protein homeostasis, inflammation, therapeutics

## Abstract

Proteinopathies involve the abnormal accumulation of specific proteins. Maintaining the balance of the proteome is a finely regulated process managed by a complex network of cellular machinery responsible for protein synthesis, folding, and degradation. However, stress and ageing can disrupt this balance, leading to widespread protein aggregation. Currently, several therapies targeting protein aggregation are in clinical trials for ALS. These approaches mainly focus on two strategies: addressing proteins that are prone to aggregation due to mutations and targeting the cellular mechanisms that maintain protein homeostasis to prevent aggregation. This review will cover these emerging drugs. Advances in ALS research not only offer hope for better outcomes for ALS patients but also provide valuable insights and methodologies that can benefit the broader field of neurodegenerative disease drug discovery.

## 1. Hypothesis of Widespread Aggregation in Proteinopathies

Proteinopathies, such as amyotrophic lateral sclerosis (ALS), Alzheimer’s disease (AD), and Parkinson’s disease (PD), are characterized at the molecular level by the presence of distinctive protein aggregates. These include TAR DNA-binding protein 43 (TDP-43) in ALS, amyloid-β (Aβ) and tau in AD, and α-synuclein in PD [1].

Protein homeostasis, or proteostasis, is the state in which a living organism’s proteome is in functional balance. This balance must be carefully regulated within individual cells, tissues, and organs. Maintaining proteostasis involves a complex network of cellular factors, including the machinery for protein synthesis, folding, and degradation. To ensure this balance, evolution has developed a robust system capable of maintaining proteome functionality [2]. However, under conditions of cellular stress and aging, the protein homeostasis system becomes increasingly impaired. This failure places additional stress on the cellular environment, burdening the proteostasis machinery and further boosting aggregation by interfering with protein folding, clearance, and other critical cellular processes, eventually failing to prevent aggregation [3,4,5,6,7]. This is evident in every neurodegenerative disease, where the main protein deposits are often associated with deposits of other proteins [3,4,5].

Protein condensation is also involved in age-related disorders such as ALS, particularly in the liquid–liquid phase separation phenomena that form membraneless organelles and granules [8]. Stress granules, for example, require autophagy for clearance [9], and the efficacy of autophagy decreases with age [10]. Additionally, chaperones play a crucial role in maintaining stress granule fluidity and their clearance [11]. These findings indicate that the failure to properly control these assemblies may lead to pathological aggregation [8]. Thus, in vivo, aggregation and aberrant condensation are inhibited by the proteostasis machinery.

Recent investigations have explored the decline of the proteostasis machinery as a potential biomarker for neurodegenerative diseases. Notably, protein aggregates can be detected in the biofluids of patients with ALS and AD [12,13]. In ALS, the presence of these aggregates in patient biofluids is particularly significant. Research has shown that the loss of TDP-43 function in ALS leads to erroneous splicing and the generation of transcripts containing cryptic exons [14]. Some of these mis-spliced transcripts are translated on the ribosomes, producing de novo proteins found in the cerebrospinal fluid (CSF) of ALS patients. These newly generated proteins contain abnormal sequences, making them more prone to misfolding, forming aberrant aggregates, and interacting abnormally with other proteins [14].

Protein aggregates, along with chaperones such as heat shock proteins that cells produce to maintain protein homeostasis, have been identified as damage-associated molecular patterns (DAMPs) capable of activating the inflammatory response [15]. Throughout the course of the disease, these DAMPs trigger the release of inflammatory mediators. Persistent inflammation can further promote protein aggregation and exacerbate existing proteostasis imbalances, creating a vicious cycle that perpetuates pathology in vulnerable cells, including motor neurons.

While the innate immune response plays a crucial role in the onset and progression of neurodegenerative diseases, emerging research suggests that the adaptive immune response and autoimmunity are also key in managing ALS [16]. Autoimmune responses have been observed targeting protein aggregates, potentially aiming to remove them [17]. Notably, the therapies currently approved for AD are antibody-based treatments designed to reduce the burden of protein aggregates [18,19]. In AD, ineffective clearance of protein aggregates has been linked to triggering immune system activation [20].

Consequently, therapeutic strategies that reduce protein aggregation or modulate the immune response are likely to benefit conditions like ALS and other diseases associated with the accumulation of protein aggregates [21].

## 2. Several Proteins Are Aggregated in ALS

To date, self-aggregating properties of protein domains and altered RNA granule formation have been suggested as contributing factors to protein aggregation in ALS. The precise pathological effects of protein aggregation remain largely unknown, but experimental evidence suggests both gain- and loss-of-function mechanisms [22]. In ALS, several protein aggregates are commonly found, including TDP-43, superoxide dismutase 1 (SOD1), fused in sarcoma (FUS), ataxin-2, optineurin, ubiquilin-2, and the translational product of chromosome 9 open reading frame 72 (C9ORF72). Mutations in the genes encoding these proteins are found in a subgroup of ALS patients and are associated with familial cases, indicating a causal relationship with the disease. Additionally, these protein aggregates are often detected in individuals with sporadic ALS, hence without these specific mutations, and in other neurodegenerative disorders [22].

In this section, we will review the aggregating proteins that are currently the target of therapeutic approaches for ALS.

SOD1 was the first protein identified to aggregate in familial ALS cases with a mutation in the Sod1 gene [23]. Mutations in this gene account for up to 2% of all ALS cases, with more than 200 different mutations identified [24]. The disease’s severity, progression rate, and prognosis vary and may be specific to the mutation type. The exact mechanisms by which SOD1 mutations cause motor neuron degeneration in genetic ALS are not fully understood. However, a toxic gain of function is considered the most likely mechanism [24]. Recently, the atomic structure of amyloid fibrils from full-length human SOD1 has been determined using cryo-electron microscopy (cryo-EM) [25].

Cytoplasmic inclusions of TDP-43 are found in the brains and/or spinal cords of approximately 97% of ALS cases, 45% of frontotemporal dementia (FTD) cases, and 40% of AD cases [14]. Mutations in TARDBP, the gene encoding TDP-43, cause familial forms of FTD and ALS, highlighting the central role of TDP-43 in disease pathogenesis. Mislocalization of TDP-43 involves both its clearance from the nucleus and the formation of cytosolic aggregates [14,26]. Nuclear clearance of TDP-43 can occur before symptom onset and precedes the formation of cytosolic aggregates, suggesting that the loss of normal TDP-43 function is an early disease mechanism [27]. Cryo-EM has been used to determine the structures of aggregated TDP-43 in individuals with ALS and frontotemporal lobar degeneration (FTLD), revealing an amyloid-like filament structure [28,29].

Mutations in the FUS gene account for 4% of familial ALS cases and 1% of sporadic ALS cases, and are often associated with young-onset disease [30]. Pathological examination of post-mortem tissue from FUS mutation carriers shows predominant degeneration of lower motor neurons with FUS-positive cytoplasmic inclusions [31]. FUS is a nuclear protein that undergoes liquid–liquid phase separation in response to stress and DNA damage. Dysregulation of this dynamic phase separation leads to the formation of pathological fibrils. The structure of a cytotoxic amyloid fibril formed by the low-complexity domain of FUS has been resolved using cryo-EM [32].

The expansion of a GGGGCC hexanucleotide repeat in the first intron of the C9orf72 coding region is the most common cause of FTLD/ALS linked to mutation. In the wild-type form, the gene harbors fewer than 25 repeats. However, in disease cases, the repeat region can extend to several hundred or thousand repeats. Although extended repeat lengths can also be detected in control cases, these expansions are strongly associated with FTLD/ALS [33,34]. Characteristic intracellular inclusions of misfolded proteins define C9orf72 pathology. These inclusions contain poly-(Gly-Ala) and, to a lesser extent, poly-(Gly-Pro) and poly-(Gly-Arg) dipeptide-repeat proteins. These proteins are presumably generated by non-ATG-initiated translation from the expanded GGGGCC repeat in three reading frames [35].

Ataxin-2 harbors a polyglutamine (polyQ) tract and belongs to a family of polyQ disease proteins, which includes the Huntington’s disease protein, huntingtin. Ataxin-2 functions in mRNA polyadenylation, stress granule formation, polyribosome assembly, and miRNA synthesis. It contains intrinsically disordered regions that facilitate the transition from a liquid-like state to solid, β-sheet-rich amyloid fibrils in vitro, particularly at high concentrations achieved in the liquid phase [36,37]. Extended polyQ repeats in ataxin-2 are associated with ALS. While ataxin-2 normally harbors 21 or 22 polyQ repeats, polyQ lengths between 27 and 33 are associated with ALS. The spinal cord tissue of sporadic ALS patients shows an increased cytoplasmic accumulation of ataxin-2 compared to controls [38]. Furthermore, studies in yeast and fly models have found that ataxin-2 promotes the aggregation and toxicity of the TDP-43 protein [38]. The pathological effect of extended ataxin-2 polyQ repeats is likely due to a gain-of-function mechanism, as lowering levels of ataxin-2 improves motor performance in mouse models [39].

## 3. Therapies Targeting Protein Aggregation in Clinical Trial for ALS

Currently, several therapies targeting protein aggregation are being tested in clinical trials for ALS. As shown in Figure 1, these therapies generally take two approaches: one targets proteins that are prone to aggregation when mutated, while the other focuses on hubs involved in maintaining protein homeostasis to prevent aggregation.

Additionally, some drugs are being investigated for their potential to modulate the immune response, which, as mentioned earlier, can result from protein aggregation. Zhang and colleagues have reviewed these drugs [40].

Table 1 provides a comprehensive list of drugs targeting protein aggregation that are currently in clinical trials. It includes details on the type of therapeutic, the target, the sponsor, the clinical trial number, and relevant references. One notable drug, Tofersen, has received approval for the treatment of SOD1-ALS in the USA and Europe. On the other hand, other drugs listed in Table 2 have not demonstrated efficacy in clinical trials and have consequently been discontinued.

In the following sections, we will cover these drugs in more detail, including those that have failed, as they offer valuable insights for future treatments. We will discuss their mechanisms of action, the scientific principles behind their development, and we will review the stages of clinical trials each drug has undergone.

### 3.1. BIIB067/Tofersen

Tofersen is an antisense oligonucleotide (ASO) that targets the mRNA of SOD1. By degrading this mRNA, tofersen prevents the synthesis of the SOD1 protein, reducing its levels. This drug was specifically developed for ALS cases associated with SOD1 mutations, which account for about 20% of familial ALS and 2% of all ALS cases. The FDA approved tofersen in April 2023, and the EMA followed with approval in May 2024 under the brand name Qalsody™.

A multiple-dosing clinical trial for tofersen began in 2016 and concluded in 2019. The results showed that ASO treatment was generally safe, with most adverse events being mild to moderate. By day 85, mutant SOD1 protein levels in cerebrospinal fluid (CSF) decreased by 3% in the low-dose group and 36% in the high-dose group compared to the placebo [41].

Although this trial was not primarily designed to measure efficacy, exploratory outcomes included changes in the Amyotrophic Lateral Sclerosis Functional Rating Scale-Revised (ALSFRS-R; 12 items across four domains of function are measured, each scored on a scale from 0 to 4, with higher scores indicating better function), respiratory function, muscle strength, and neurofilament concentrations in blood and CSF. The highest-dose group showed a slower decline on the ALSFRS-R, losing 1.19 points at 85 days compared to 5.63 points in the placebo group. Additionally, concentrations of neurofilament heavy and light chains in CSF and blood decreased from baseline to 12 weeks in the highest-dose group [41].

In March 2019, a Phase 3 efficacy arm named VALOR was added to the ongoing trial. This study enrolled 108 participants with SOD1-ALS, who were randomly assigned in a 2:1 ratio to receive either eight infusions of 100 mg tofersen or a placebo over 24 weeks. The primary endpoint was the change in the ALSFRS-R score from baseline to week 28 in participants with fast-progressing disease. Secondary endpoints included changes in SOD1 protein concentration in CSF, neurofilament light chains in plasma, respiratory function, and muscle strength. Although the VALOR trial did not meet its primary endpoint of slowing the ALSFRS-R decline at week 28 with tofersen not being able to improve clinical endpoints, the drug did reduce SOD1 concentrations in CSF and neurofilament light chains in plasma over the 28 weeks [42].

In April 2023, the FDA granted tofersen accelerated approval after an advisory committee unanimously agreed that the reduction in neurofilament light chains (NfL) observed in the VALOR trial likely indicated a clinical benefit. According to the FDA regulations, the company must demonstrate efficacy through the ongoing Phase 3 Atlas trial, which will continue until 2027 [54]. In May 2023, tofersen was approved in the European Union as well by the EMA.

### 3.2. AP-101

AP-101 is a fully human IgG1 antibody that exhibits high affinity and selective binding to the misfolded SOD1 protein. The presence of misfolded SOD1 can be detected in the majority of ALS patients, indicating that SOD1 is a common pathogenic factor in both familial and sporadic forms of ALS. Research conducted in transgenic mouse models of ALS demonstrated that a murine version of AP-101 was effective in reducing the loss of spinal cord motor neurons and extending overall survival [43].

AP-101 has successfully completed a Phase 1 clinical trial (NCT03981536), which established that the treatment posed no safety concerns and was well tolerated across all tested doses. Building on these promising results, AP-101 is currently being evaluated in a larger Phase 2a study (NCT05039099). This study aims to further assess the safety and tolerability of the treatment. Additionally, it seeks to investigate pharmacodynamic markers and pharmacokinetics of AP-101 in patients with both familial and sporadic ALS.

### 3.3. AMT-162

AMT-162 is an investigational gene therapy designed to encode an artificial microRNA (miRNA) that specifically targets the SOD1 gene. This therapy is currently undergoing a Phase 1/2 clinical trial (NCT06100276) involving participants with SOD1-ALS. The primary objectives of this study are to evaluate the safety and tolerability of intrathecally administered AMT-162, as well as to explore its preliminary efficacy. The underlying hypothesis of this gene therapy is that it will silence the expression of the mutant cytosolic SOD1 gene, thereby slowing or halting the progression of ALS associated with this genetic mutation.

### 3.4. ION363/Jacifusen

ION363 is an ASO designed to target and reduce the production of a mutated, neurotoxic form of the FUS protein. Preclinical testing of ION363 was conducted in FUS knock-in mouse lines engineered to express ALS-associated mutant forms of the FUS protein, specifically FUS-P525L and FUS-ΔEX14. These mutant mice exhibit age-dependent motor neuron loss due to a dose-dependent toxic gain of function, which is associated with the insolubility of FUS and other related RNA-binding proteins [44].

The results from these preclinical studies were promising. ION363 efficiently silenced the FUS gene, resulting in a significant reduction of FUS protein levels in both the brain and spinal cord after birth. This reduction delayed the degeneration of motor neurons in the mutant mice, indicating a potential therapeutic effect for ALS patients with FUS mutations [44].

Building on these promising preclinical results, a first-in-human study was conducted. In this study, a patient with ALS-FUS, carrying the FUS-P525L mutation, received repeated intrathecal administrations of ION363. The treatment led to a marked suppression of FUS expression in the patient’s brain and spinal cord. Additionally, there was a significant reduction in FUS aggregates, which are a pathological hallmark of this form of ALS. These findings suggest that ION363 could potentially modify the course of the disease by targeting the underlying genetic mutation [44].

To further investigate the efficacy and safety of ION363, a Phase 3 clinical trial named FUSION (NCT04768972) was initiated in June 2021. This trial is expected to run through June 2026 and has enrolled 77 patients worldwide. The primary objective of the FUSION trial is to evaluate functional changes in participants using the revised ALS-FRS and to assess the duration of life free from mechanical ventilation. Secondary objectives include assessing quality of life, lung and muscle function, overall survival, and changes in the levels of the cerebrospinal fluid biomarker neurofilament light chain.

### 3.5. Colchicine

Colchicine is an alkaloid derived from the plant Colchicum autumnale, which belongs to the Lily family. Approved by the FDA in 1961, this drug is used to treat gout flares, familial Mediterranean fever, and to help prevent major cardiovascular events.

The rationale for investigating colchicine in the context of ALS relies on its effect on enhancing the expression of heat shock protein B8 (HSPB8) and several autophagy players, including the key autophagy gene TFEB [55]. Disruptions in the ubiquitin–proteasome system and autophagy are central to ALS pathology. The molecular chaperone HSPB8 plays a crucial role in recognizing and promoting the removal of misfolded mutant SOD1 and TDP-43 fragments from ALS motor neurons through autophagy [56,57]. Additionally, HSPB8, along with BAG3 and HSP70, is involved in maintaining the so-called ‘granulostasis’ mechanism, a quality control mechanism that prevents dynamic stress granules from converting into aggregation-prone assemblies [58].

This drug was evaluated in a Phase 2 trial (NCT03693781) aimed at assessing its effects on disease progression as defined by changes in the revised ALSFRS-R, as well as its safety and tolerability in ALS patients [45]. The trial has concluded, and the results were recently published [46]. The findings indicate that colchicine did not demonstrate efficacy in this small trial. Approximately one-third of patients treated with a low dose of colchicine met the study’s primary endpoint, showing no more than a 4-point decline in ALSFRS-R scores after 30 weeks. In comparison, 13.3% of the placebo group reached this outcome, suggesting slower disability progression with colchicine, though this difference was not statistically significant. Only 7.1% of patients on the higher dose showed a positive response. There was also a trend toward longer survival and better-preserved respiratory function in the low-dose group, although these findings were again not statistically significant. Secondary outcomes, including biomarkers of neurodegeneration, inflammation, and autophagy, revealed no significant differences between the groups.

Although the frequency of positive responses was similar across groups, the monthly decline in ALSFRS-R scores was significantly slower in the low-dose group compared to the placebo group, and this effect persisted post-treatment. Notably, toxic TDP-43 levels increased in the placebo group but decreased with colchicine, with a significant difference observed at 30 weeks between placebo and colchicine-treated patients. Adverse events were comparable across all groups, with no statistically significant differences.

In conclusion, colchicine appeared safe for patients with ALS. However, the researchers cautioned that these findings should be interpreted carefully due to the small sample size and stated that further trials are needed to accurately assess the potential efficacy of colchicine in ALS treatment [46].

### 3.6. Guanabenz

Guanabenz is a drug that has been classified as an orphan drug (ORPHA178067) and was granted approval by the European Medicines Agency in 2009 for the treatment of traumatic spinal cord injury. Originally, Guanabenz was used as an antihypertensive medication due to its role as an agonist of the α-2-adrenergic receptor [48,59]. However, recent research has uncovered additional benefits of Guanabenz beyond its initial use. Specifically, it has been found to offer cellular protection against protein misfolding stress by inhibiting the dephosphorylation of eukaryotic translation initiation factor 2α (eIF2α) [60]. Guanabenz achieves this by binding to a regulatory subunit of protein phosphatase 1, known as PPP1R15A. This binding action prolongs the phosphorylation of eIF2α in stressed cells, thereby regulating protein production rates to levels that can be managed by available chaperones [60].

In a Phase 2 clinical study conducted in Italy (EudraCT Number: 2014-005367-32), Guanabenz was tested on patients with ALS to evaluate its clinical efficacy, safety, tolerability, and its impact on biomarkers of disease progression. The study involved administering Guanabenz to ALS patients over a period of 6 months. The results, published in 2021, indicated that Guanabenz successfully met the primary hypothesis of non-futility. This was evidenced by a significantly lower proportion of patients progressing to more advanced stages of the disease at 6 months compared to what was expected under the non-futility hypothesis. Additionally, there was a significant reduction in the median rate of change in the total revised ALSFRS score, particularly among patients with bulbar onset of the disease [47].

Despite these positive outcomes, the study also revealed some challenges. A higher proportion of patients in the Guanabenz treatment groups experienced at least one adverse event compared to those in the placebo group. The incidence of drug-related side effects increased with higher doses of Guanabenz, and the highest dose group had a significantly higher dropout rate. These findings suggest that while Guanabenz’s potential to modulate the unfolded protein response pathway in ALS patients is promising, the drug’s α-2 adrenergic-related side effects resulted in poor tolerability among patients [47]. Optimizing the dosage to ensure that the benefits outweigh the side effects is crucial for developing an effective treatment for ALS.

### 3.7. DNL343

DNL343 is a brain-permeant small molecule that inhibits the cell’s unfolded protein response and restores protein synthesis. It acts as an activator of the eukaryotic initiation factor eIF2B. Normally, eIF2B, along with its eIF2A subunit, initiates mRNA translation, and its activity is regulated by the integrated stress response (ISR), a conserved signaling pathway activated by various cellular stresses, including protein misfolding [61].

Under cellular stress conditions, the ISR inhibits eIF2B, pausing protein synthesis. Loss-of-function mutations in eIF2B that impair protein translation can cause a progressive neurodegenerative syndrome affecting the brain and spinal cord, known as vanishing white matter disease [62]. Restoring eIF2B activity using PKR-like ER kinase (PERK) inhibitors or genetic approaches has been shown to protect against neurodegeneration in preclinical models of prion disease, frontotemporal dementia, and ALS [63].

In immortalized cells and iPSC-derived neurons expressing ALS-linked mutations exposed to stressors, DNL343 inhibits the ISR and both prevents and reverses stress granule formation [49]. The high potency, selectivity, and pharmacokinetic profile of DNL343 across preclinical species make it an ideal tool for studying ISR inhibition in vivo [64].

Developed by Denali Therapeutics, DNL343 has completed a Phase 1 clinical trial in healthy participants (NCT04268784) and is now undergoing clinical development for ALS (NCT05006352, NCT05842941). The Phase 2/3 study began in May 2023 and will continue through July and August 2025, when the results are expected. The primary outcomes are disease progression on the revised ALS-FRS and mortality. Secondary outcomes include muscle strength, respiratory function, and serum neurofilament light chain levels.

### 3.8. ABBV-CLS-7262

ABBV-CLS-7262, another small molecule, is being evaluated by AbbVie and Calico Life Sciences as an activator of eIF2B. This compound is currently in a Phase 2/3 trial that started in March 2023 and will run until September 2024. The study completion and the results are expected in October 2024. This study is part of the Healey ALS platform trial, a multicenter clinical collaboration testing different interventions in parallel, including DNL343, against a common placebo group in ALS patients. Similar to DNL343, the primary outcomes for ABBV-CLS-7262 are disease progression on the revised ALS-FRS and mortality. Secondary outcomes include muscle strength, respiratory function, and serum neurofilament light chain levels.

Both ABBV-CLS-7262 and DNL343 are derivatives of the integrated stress response inhibitor ISRIB, developed by Peter Walter’s laboratory at the University of California, San Francisco [65]. ISRIB has been reported to have neuroprotective effects, including protecting neurons from SOD1 toxicity in a cellular model of ALS [66].

### 3.9. Metformin

Beyond these ISR-specific drugs, compounds like metformin, which have a broad effect on metabolism, can also influence the ISR [67]. Metformin is an FDA-approved drug currently used to treat type 2 diabetes by controlling blood sugar levels [68]. Research has shown that gene expansion mutations can produce proteins synthesized from multiple reading frames without the AUG starting codon through a process called repeat-associated non-AUG (RAN) translation [69]. This phenomenon has been reported in C9orf72-related ALS and frontotemporal dementia, where the C9orf72 GGGGCC repeat is translated into aggregating dipeptide-repeat proteins [35,69].

In C9orf72 mouse models of ALS/FTD, metformin reduces the levels of RAN proteins produced due to the C9orf72 mutation, thereby improving disease features [67]. Metformin works by blocking the double-stranded RNA-dependent protein kinase (PKR) pathway, which is essential for producing RAN proteins. PKR is a serine/threonine kinase that phosphorylates the translation initiation factor eIF2α, impairing the initiation of translation for most proteins [67].

Metformin is now being investigated as a potential treatment for ALS in patients with the C9orf72 mutation. Given its established safety and tolerability, a Phase 1 study for ALS was not required. Currently, a Phase 2 study is assessing the safety, tolerability, and efficacy of metformin in ALS patients (NCT04220021). The results are expected by the end of the 2024.

### 3.10. Ambroxol

Ambroxol is an expectorant commonly used to treat respiratory diseases characterized by excessive mucus secretion. This drug effectively reduces the viscosity of mucus, making it easier to expel. In addition to its mucolytic properties, Ambroxol has antioxidant, anti-inflammatory, and anesthetic effects [70].

A screening of 1040 FDA-approved drugs identified Ambroxol as a potent chaperone for Glucocerebrosidase (GCase), an enzyme responsible for breaking down glucocerebroside into glucose and ceramide in the lysosome [71]. Mutations in the GBA1 gene, which encodes GCase, significantly increase the risk of developing Parkinson’s disease. Homozygous mutations in GBA1 result in a deficiency of GCase, leading to Gaucher’s disease. The malfunction of GCase impairs the lysosomal degradation system, affecting autophagy and the unfolded protein response pathways in both Parkinson’s and Gaucher’s diseases [72,73].

Ambroxol has demonstrated the ability to improve motor functions and extend survival in a SOD1-ALS mouse model [74]. Currently, it is undergoing a Phase 2 trial called AMBALS (NCT05959850), which aims to assess the long-term safety and effectiveness of Ambroxol for people with ALS. The completion of the study is expected in December 2024.

### 3.11. WVE-004

WVE-004 is an ASO designed to target the chromosome 9 open reading frame 72 (C9orf72) gene mRNA. It works by mediating the degradation of mRNAs containing the hexanucleotide expansion in the C9orf72 gene. In laboratory studies, WVE-004 has been shown to reduce the levels of repeat-containing mRNA in cultured cells. When administered via injection into the CSF of mice, it achieved a significant reduction in repeat-containing mRNA levels in the spinal cord and in the cortex, with these effects lasting up to six months. Additionally, WVE-004 decreased dipeptide repeat proteins by approximately 90% in both the spinal cord and cortex, without affecting normal protein levels [75].

In 2021, a Phase 1/2 safety study (NCT04931862) began to evaluate WVE-004 in patients with ALS or FTD associated with the C9orf72 mutation. However, in May 2023, the development of the drug was halted. Despite observing a 50% reduction in CSF poly(GP) levels with several of the trialed doses, there were no significant improvements in exploratory clinical outcomes after 24 weeks. Furthermore, changes in poly(GP) levels did not correlate with clinical improvements in individual participants.

### 3.12. BIIB078

BIIB078 is an ASO targeting the C9orf72 mRNA, specifically mediating the degradation of mRNAs containing the hexanucleotide expansion. Preclinical studies using patient-derived cell lines demonstrated that C9orf72 ASOs could mitigate RNA toxicity [76]. In mouse models expressing the C9orf72 expansion, a single intraventricular dose of the ASO targeting the hexanucleotide repeats successfully reduced toxic RNA and dipeptide aggregates, and also alleviated behavioral and cognitive deficits [77].

In 2022, based on the results of a Phase 1 study, which included secondary and exploratory findings showing no reduction in neurofilament levels and no improvement in clinical outcomes compared to the placebo group, Biogen decided to discontinue the clinical development of BIIB078 [52]. Despite this decision, the results from this study will contribute valuable insights to the understanding of the complex pathobiology of C9orf72-associated ALS.

### 3.13. BIIB105

This drug is an ASO that targets and degrades ataxin-2 mRNA, reducing ataxin-2 protein levels. Since ataxin-2 exacerbates TDP-43 protein aggregation [38], lowering its levels could potentially help many ALS patients. This differs from other ASOs in development, such as tofersen and ION363, which target specific genetic mutations in SOD1 and FUS and are only relevant to a minority of patients. In preclinical studies, an ataxin-2 ASO developed by Ionis improved survival and motor function in a TDP-43 mouse model [39].

A Phase 1 trial of BIIB105 began in 2020 and was upgraded to Phase 1/2 in 2022 under the name ALSpire. The trial aimed to evaluate efficacy through various endpoints, including biomarkers and clinical measures. However, in May 2024, Biogen and Ionis announced the discontinuation of BIIB105 due to negative topline results. While BIIB105 reduced ataxin-2 levels in cerebrospinal fluid, it did not impact neurofilament light chain levels or clinical measures of function, breathing, and strength.

### 3.14. Arimoclomol

Arimoclomol is a co-inducer of heat shock protein 70 (HSP70) known for its neuroprotective effects in animal models of ALS. It works through several mechanisms, including the clearance of protein aggregates.

The safety and efficacy of arimoclomol in ALS patients were evaluated through the ORARIALS-01 Phase 3 trial (NCT03491462). The trial found that arimoclomol did not show improvement in efficacy outcomes compared to placebo, and mortality rates were similar between the treatment groups [53]. Although current biomarker data from this trial do not rule out future strategies targeting the HSP response, safety concerns suggest that a higher dose of arimoclomol might not be feasible [53].

## 4. Conclusions

Targeting protein aggregation in ALS represents a promising therapeutic strategy, as demonstrated by the number of drugs with this mechanism of action currently undergoing clinical trials. In addition to impaired proteostasis, several other mechanisms of ALS pathogenesis have been proposed and are being explored as therapeutic targets. These include altered RNA metabolism, nucleocytoplasmic transport defects, impaired DNA repair, mitochondrial dysfunction and oxidative stress, axonal and vesicular transport defects, neuroinflammation, excitotoxicity, and oligodendrocyte dysfunction [78]. Notably, the approval of tofersen by the FDA and EMA, a disease-modifying drug that targets protein aggregation, highlights the potential of this approach. However, four drugs aimed at similar mechanisms have been discontinued due to a lack of efficacy. These failures do not indicate that the approach is flawed, but rather reflect the complex and heterogeneous nature of ALS, alongside the difficulties in translating preclinical successes into human treatments. Despite these setbacks, the lessons learned from the failure of anti-aggregation drugs provide valuable insights for the ongoing development of effective ALS treatments.

Translating preclinical findings into human efficacy remains a significant challenge. One major hurdle is optimizing dosing and managing side effects that may outweigh the benefits, raising concerns about safety and tolerability. In the case of ALS, clinical trials face an additional challenge: the complexity of the disease’s pathology, which can dilute therapeutic effects. ALS is a multifactorial disease characterized by substantial variability among patients [79]. This variability leads to wide-ranging responses to treatment, further complicating efforts to develop effective therapies. A deeper understanding of the disease’s onset is crucial, as early intervention is key. However, ALS presents differently in each patient, with variations in symptoms, progression rates, and underlying genetic factors [78,79]. This is where the search for molecular biomarkers becomes critical to improving the chances of therapeutic success. Biomarkers not only provide insights into the onset and progression of the disease but also enable early diagnosis and precise patient stratification, which is essential for optimizing clinical trial design. In AD, for example, the time course of various biomarkers has not only facilitated early diagnosis before clinical symptoms emerge but also allowed the classification of distinct forms of the disease [80]. This knowledge has illuminated the temporal development of biochemical and histological changes, offering valuable insights for drug development. Similarly, advancing biomarker research in ALS holds the potential to transform our approach to treatment, enabling earlier interventions and more personalized therapies that could finally make a meaningful difference for patients.

The complexity of ALS poses significant challenges in developing research models that accurately mimic the human disease, which in turn hampers the discovery of potential therapeutics. Translational research models are essential for bridging the gap between preclinical findings and the clinical applications of drug treatments. This issue is well illustrated by the clinical failures of AD drugs, which have largely been attributed to the limitations of current preclinical studies. These studies often rely heavily on transgenic mouse models that frequently fail to replicate the complexities of the human condition [81]. A promising alternative to these animal models is the use of induced pluripotent stem cell (iPSC)-derived systems [82]. These human-derived cells offer a valuable platform for studying disease mechanisms and drug responses in a more human-relevant context. The recent enactment of the FDA Modernization Act 2.0 [83] further supports this shift in preclinical research by encouraging the use of alternative methods, reducing reliance on animal models, and promoting more accurate translational research [84].

In summary, targeting protein aggregation in ALS is a promising therapeutic strategy. This is evidenced by the therapeutics currently in clinical trials that aim to prevent or reduce protein aggregation, as reviewed here. The development of disease-modifying therapies for major neurodegenerative disorders has proven challenging. However, ALS is one of the few neurodegenerative diseases for which such therapies have been approved. Advances in biomarker development and preclinical models are crucial for translating these approaches into effective treatments. Progress in ALS research not only holds promise for improving outcomes for ALS patients but also provides valuable insights and methodologies that can be applied to the broader field of neurodegenerative disease drug discovery.

## Figures and Tables

**Figure 1 biomolecules-14-01324-f001:**
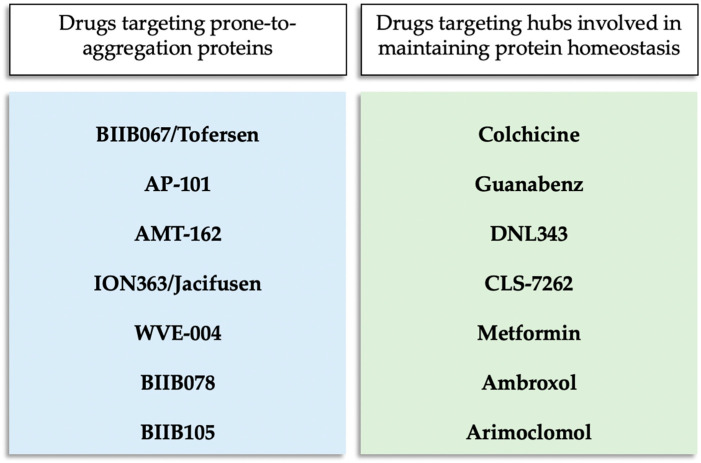
Drugs targeting protein aggregation in ALS categorized according to their mode of action.

**Table 1 biomolecules-14-01324-t001:** Therapeutics approaches targeting protein aggregation in ALS in clinical trial.

Name	Type	Target	Sponsor	Clinical Trial	References
BIIB067/Tofersen	ASO	SOD1	Biogen	NCT02623699; NCT03070119	[41,42]
AP-101	antibody	SOD1	AL-S Pharma AG	NCT05039099	[43]
AMT-162	miRNA	SOD1	UniQure Biopharma BV	NCT06100276	-
ION363/Jacifusen	ASO	FUS	Ionis Pharmaceuticals	NCT04768972	[44]
Colchicine	Small molecule	HSPB8	Azienda Ospedaliero-Universitaria di Modena	NCT03693781	[45,46]
Guanabenz	Small molecule	PPP1R15A	Fondazione IRCCS Istituto Neurologico Carlo Besta	EudraCT 2017-001042-10	[47,48]
DNL343	Small molecule	eIF2B	Denali Therapeutics/ Massachusetts General Hospital	NCT05006352; NCT05842941	[49,50]
CLS-7262	Small molecule	eIF2B	AbbVie, Calico Life Sciences/Massachusetts General Hospital	NCT05740813	[51]
Metformin	Small molecule	PKR pathway	University of Florida	NCT04220021	-
Ambroxol	Small molecule	GBA2	The Florey Institute of Neuroscience and Mental Health	NCT05959850	-

**Table 2 biomolecules-14-01324-t002:** Discontinued therapeutics approaches targeting protein aggregation in ALS.

Name	Type	Target	Sponsor	Clinical Trial	References
WVE-004	ASO	C9orf72	Wave Life Sciences	NCT04931862	-
BIIB078	ASO	C9orf72	Biogen	NCT03626012	[52]
BIIB105	ASO	ataxin-2	Biogen	NCT04494256	-
Arimoclomol	Small molecule	HSP70	Orphazyme	NCT03491462	[53]

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
