# Peer review of "Targeting Protein Aggregation in ALS"

_biomolecules, 2024, doi:10.3390/biom14101324_

Round 1
Reviewer 1 Report
Comments and Suggestions for Authors
Dear Editor
I have reviewed the article entitled “Targeting Protein Aggregation in ALS” by
Michele Perni and Benedetta Mannini. Although the review is well-written and the approaches suggested for targeting ALS are suitably chosen, I have concerns about the following aspects, which the authors can address and improve the review. Also, integrating some of these suggestions may help remove the bias.
In the quest to tackle Amyotrophic Lateral Sclerosis (ALS) through the reduction of protein aggregation, several drugs, which authors list in the review, have been explored with varying degrees of promise. Here's a brief overview of each drug and its challenges in clinical trials. I am highlighting and reiterating these aspects to remove the bias that the authors have not addressed well. The authors must note these seriously and address the challenges and varying degrees of promise and toxicity issues.
1. Colchicine: Traditionally used as an anti-inflammatory and gout treatment, colchicine has been tested for ALS due to its potential effects on microtubules and protein aggregation. However, its effectiveness has been limited by issues such as inadequate drug delivery to the central nervous system (CNS) and potential toxicity at therapeutic doses. The lack of substantial clinical benefit observed in trials has curtailed the development of ALS. One notable fact is that Colchicine can cause chromosomal aberrations. The reference 43 that the authors have cited dates back to 2019, meaning the results are no longer awaited, and the trial didn’t work or was abandoned due to the abovementioned issues. Authors should bring about this in the text.
2. Guanabenz: This alpha-2 adrenergic agonist has shown potential in preclinical models for activating the unfolded protein response (UPR), which can help manage protein misfolding and aggregation. Despite promising preclinical data, clinical trials have not demonstrated a consistent benefit in ALS patients. Challenges include optimizing dosing and addressing side effects that might outweigh the benefits.
3. DNL343: An experimental drug developed to target protein aggregation and misfolding, DNL343 aims to enhance the cellular degradation pathways. Although preclinical studies showed promise, translating these findings into human efficacy has been difficult. Trials have faced issues related to drug tolerability, dosing, and the complexity of ALS pathology, which may dilute the therapeutic effects.
4. CLS-7262: This drug targets protein misfolding and aggregation and has been studied for its potential neuroprotective effects. However, its clinical trials have not shown significant improvements in ALS outcomes. Challenges include the multifactorial nature of ALS and the need for a more precise understanding of how the drug interacts with disease mechanisms. I want the authors to read the review by Matt Keon, “Destination ALS” (Keon M, Musrie B, Dinger M, Brennan SE, Santos J, Saksena NK. Destination Amyotrophic Lateral Sclerosis. Front Neurol. 2021 Mar 29;12:596006. doi: 10.3389/fneur.2021.596006. PMID: 33854469; PMCID: PMC8039771). This review brings about a philosophical issue of catching the disease before it becomes multifactorial. Why some of these drugs do not work because substantial time has been allowed for the disease, in this case, ALS, to become multifactorial, please address this in your conclusion section.
5. Metformin: Known primarily for its use in diabetes, metformin has been investigated for its potential neuroprotective effects and its ability to affect protein aggregation through various cellular pathways. While some studies suggest metformin may have neuroprotective properties, clinical trials in ALS have yielded mixed results. Variability in patient responses and the complexity of ALS likely contribute to these outcomes. This is a significant and the most dampening issue for clinical trials: patient variability. The authors should address the problem of sporadic and non-sporadic ALS, along with patient variability, as well as the drug efficacy at the time of the disease from its onset.
6. Ambroxol: This drug, typically used as a mucolytic agent, has shown promise in preclinical studies for its potential to enhance lysosomal function and reduce protein aggregation. Despite some positive results in early-phase trials, the clinical benefits observed have not been consistent. Challenges include optimizing dosing and addressing variability in patient responses.
7. Arimoclomol: This drug aims to enhance the heat shock response, which helps manage protein misfolding and aggregation. Initial trials were promising, but more extensive and rigorous studies have not consistently shown significant clinical improvements. Issues such as dosing, patient variability, and the multifaceted nature of ALS contribute to the challenges in demonstrating apparent efficacy.
I want the authors to bring in a section that explains briefly
Why These Approaches Haven’t Worked Effectively to date in ALS trials. This section is essential as we must remove the bias and keep the reader well-informed.
• Complexity of ALS: ALS is a heterogeneous disease with diverse pathological processes. Treatments targeting specific mechanisms may not address all aspects of the disease, leading to mixed or limited results.
• Drug Delivery Challenges: Many drugs face difficulties in effectively reaching the CNS in adequate concentrations, impacting their efficacy in treating neurodegenerative diseases like ALS.
• Variability in Patient Response: Differences in genetic background, disease stage, and individual responses can affect how well a drug works, making it challenging to achieve consistent results across clinical trials.
• Side Effects and Toxicity: Balancing drug efficacy with manageable side effects is crucial. Some drugs have shown promise in preclinical studies but have encountered problems with safety or tolerability in humans.
• Inadequate Models: Animal models and cell-based systems used in preclinical research do not always fully replicate human ALS, leading to discrepancies between early results and clinical trial outcomes.
In summary, while these drugs offer potential avenues for reducing protein aggregation in ALS, translating preclinical success into meaningful clinical benefits remains challenging. Future research may need to focus on more personalized approaches, improved drug delivery methods, and a better understanding of ALS pathology to enhance treatment effectiveness.
Overall, the review is interesting and would find a wide readership as it discusses an important issue.
Author Response
Reviewer 1 Comments and Suggestions for Authors
I have reviewed the article entitled “Targeting Protein Aggregation in ALS” by Michele Perni and Benedetta Mannini. Although the review is well-written and the approaches suggested for targeting ALS are suitably chosen, I have concerns about the following aspects, which the authors can address and improve the review. Also, integrating some of these suggestions may help remove the bias.
In the quest to tackle Amyotrophic Lateral Sclerosis (ALS) through the reduction of protein aggregation, several drugs, which authors list in the review, have been explored with varying degrees of promise. Here's a brief overview of each drug and its challenges in clinical trials. I am highlighting and reiterating these aspects to remove the bias that the authors have not addressed well. The authors must note these seriously and address the challenges and varying degrees of promise and toxicity issues.
- Colchicine: Traditionally used as an anti-inflammatory and gout treatment, colchicine has been tested for ALS due to its potential effects on microtubules and protein aggregation. However, its effectiveness has been limited by issues such as inadequate drug delivery to the central nervous system (CNS) and potential toxicity at therapeutic doses. The lack of substantial clinical benefit observed in trials has curtailed the development of ALS. One notable fact is that Colchicine can cause chromosomal aberrations. The reference 43 that the authors have cited dates back to 2019, meaning the results are no longer awaited, and the trial didn’t work or was abandoned due to the abovementioned issues. Authors should bring about this in the text.
Authors’ response
We sincerely appreciate the reviewer’s thoughtful comments, which have helped us improve the clarity of the manuscript.
We agree with the reviewer’s point regarding the status of the Colchicine trial. Since our submission in August 2024, the results of the trial were indeed published in September 2024. In response, we have updated the manuscript by citing the newly available reference (now reference 46) and discussing the trial’s outcome in detail. Specifically, we have added a paragraph (lines 277-297) that addresses the trial's results, its conclusion, and the future prospects for Colchicine as a therapeutic option for ALS. All modifications have been highlighted in yellow for ease of reference.
- Guanabenz: This alpha-2 adrenergic agonist has shown potential in preclinical models for activating the unfolded protein response (UPR), which can help manage protein misfolding and aggregation. Despite promising preclinical data, clinical trials have not demonstrated a consistent benefit in ALS patients. Challenges include optimizing dosing and addressing side effects that might outweigh the benefits.
Authors’ response
We appreciate the reviewer’s valuable comments regarding the challenges in optimizing Guanabenz dosing for effective translation to clinical use. In response, we have strengthened this point by expanding the discussion in the manuscript (lines 326-328). While Guanabenz presents adverse events, it is important to highlight that the drug successfully met the primary non-futility hypothesis, as evidenced by a significantly lower proportion of patients progressing to more advanced stages of the disease at 6 months than expected under the non-futility assumption.
Additionally, based on the reviewer's insight, we have further elaborated in the conclusion section on the balance between side effects and therapeutic benefits (lines 469-471). All modifications are highlighted in yellow for ease of reference.
- DNL343: An experimental drug developed to target protein aggregation and misfolding, DNL343 aims to enhance the cellular degradation pathways. Although preclinical studies showed promise, translating these findings into human efficacy has been difficult. Trials have faced issues related to drug tolerability, dosing, and the complexity of ALS pathology, which may dilute the therapeutic effects.
Authors’ response
We thank the reviewer for pointing out the need for clarity regarding the status of the DNL343 trial. As the trial is still ongoing, we have not discussed its outcome in the manuscript. To avoid any confusion, we have now included the expected date for the trial results (lines 347-348), clarifying that the lack of discussion is due to the trial’s ongoing nature, not drug inefficacy.
Additionally, following the reviewer’s suggestion, we have expanded the conclusion to discuss key challenges, including drug tolerability, dosing, and the complexity of ALS pathology, which may affect therapeutic outcomes (lines 469-471). All modifications are highlighted in yellow for ease of reference.
- CLS-7262: This drug targets protein misfolding and aggregation and has been studied for its potential neuroprotective effects. However, its clinical trials have not shown significant improvements in ALS outcomes. Challenges include the multifactorial nature of ALS and the need for a more precise understanding of how the drug interacts with disease mechanisms. I want the authors to read the review by Matt Keon, “Destination ALS” (Keon M, Musrie B, Dinger M, Brennan SE, Santos J, Saksena NK. Destination Amyotrophic Lateral Sclerosis. Front Neurol. 2021 Mar 29;12:596006. doi: 10.3389/fneur.2021.596006. PMID: 33854469; PMCID: PMC8039771). This review brings about a philosophical issue of catching the disease before it becomes multifactorial. Why some of these drugs do not work because substantial time has been allowed for the disease, in this case, ALS, to become multifactorial, please address this in your conclusion section.
Authors’ response
We appreciate the reviewer’s insightful comments regarding CLS-7262 and for pointing out the need for clarity regarding the status of the trial results. As the trial is still ongoing, we have not discussed its outcome in the manuscript. To clarify that the lack of discussion is due to the ongoing nature of the trial rather than drug inefficacy, we have added the expected date for the trial results (lines 354-355).
Furthermore, in response to the reviewer’s suggestion, we have addressed the complexity and multifactorial nature of ALS in the conclusion section (lines 471-478). We also incorporated insights from the recommended review by Matt Keon, highlighting the need of early intervention in ALS (now cited as reference 79). All modifications are highlighted in yellow for ease of reference.
- Metformin: Known primarily for its use in diabetes, metformin has been investigated for its potential neuroprotective effects and its ability to affect protein aggregation through various cellular pathways. While some studies suggest metformin may have neuroprotective properties, clinical trials in ALS have yielded mixed results. Variability in patient responses and the complexity of ALS likely contribute to these outcomes. This is a significant and the most dampening issue for clinical trials: patient variability. The authors should address the problem of sporadic and non-sporadic ALS, along with patient variability, as well as the drug efficacy at the time of the disease from its onset.
Authors’ response
We thank the reviewer for pointing out the need for clarity regarding the status of the Metformin trial. As the trial is still ongoing, we have not discussed its outcome. To clarify that the absence of discussion is due to the ongoing nature of the trial rather than drug inefficacy, we have added the expected date for the trial results (lines 383-384).
In addition, we appreciate the reviewer’s comments regarding the complexity of patient variability in ALS. In response to the reviewer’s insights, we have addressed the issue of patient variability and drug efficacy relative to disease in the conclusion section (lines 474-478). All modifications are highlighted in yellow for ease of reference.
- Ambroxol: This drug, typically used as a mucolytic agent, has shown promise in preclinical studies for its potential to enhance lysosomal function and reduce protein aggregation. Despite some positive results in early-phase trials, the clinical benefits observed have not been consistent. Challenges include optimizing dosing and addressing variability in patient responses.
Authors’ response
We thank the reviewer for their insightful comments regarding Ambroxol. We acknowledge the need for clarity about the status of the Ambroxol trial. As the trial is still ongoing, we have not discussed its outcome in the manuscript. To clarify that the absence of discussion is due to the trial's ongoing nature and not drug inefficacy, we have added the expected date for the trial results (lines 400). Furthermore, following the reviewer’s suggestions, we have expanded the conclusion section (lines 469-478) to address challenges related to dosing and variability in patient responses. All modifications are highlighted in yellow for ease of reference.
- Arimoclomol: This drug aims to enhance the heat shock response, which helps manage protein misfolding and aggregation. Initial trials were promising, but more extensive and rigorous studies have not consistently shown significant clinical improvements. Issues such as dosing, patient variability, and the multifaceted nature of ALS contribute to the challenges in demonstrating apparent efficacy.
Authors’ response
We thank the reviewer for their insightful comments. We agree that, unfortunately, Arimoclomol failed to demonstrate efficacy in clinical trials for ALS. We appreciate the reviewer’s highlighting of the factors contributing to this outcome, such as dosing challenges, patient variability, and the multifaceted nature of ALS. These issues have been discussed in the conclusion section (lines 469-488). All modifications are highlighted in yellow for ease of reference.
I want the authors to bring in a section that explains briefly
Why These Approaches Haven’t Worked Effectively to date in ALS trials. This section is essential as we must remove the bias and keep the reader well-informed.
- Complexity of ALS: ALS is a heterogeneous disease with diverse pathological processes. Treatments targeting specific mechanisms may not address all aspects of the disease, leading to mixed or limited results.
- Drug Delivery Challenges: Many drugs face difficulties in effectively reaching the CNS in adequate concentrations, impacting their efficacy in treating neurodegenerative diseases like ALS.
- Variability in Patient Response: Differences in genetic background, disease stage, and individual responses can affect how well a drug works, making it challenging to achieve consistent results across clinical trials.
- Side Effects and Toxicity: Balancing drug efficacy with manageable side effects is crucial. Some drugs have shown promise in preclinical studies but have encountered problems with safety or tolerability in humans.
- Inadequate Models: Animal models and cell-based systems used in preclinical research do not always fully replicate human ALS, leading to discrepancies between early results and clinical trial outcomes.
In summary, while these drugs offer potential avenues for reducing protein aggregation in ALS, translating preclinical success into meaningful clinical benefits remains challenging. Future research may need to focus on more personalized approaches, improved drug delivery methods, and a better understanding of ALS pathology to enhance treatment effectiveness.
Overall, the review is interesting and would find a wide readership as it discusses an important issue.
Authors’ response
We thank the reviewer for their valuable suggestion to include a dedicated section on the challenges faced in ALS clinical trials. In response, we have added a new section in the conclusion (lines 455-501), where we discuss the complexities involved in translating preclinical findings into human trials. This section addresses the reasons why many of the drugs, included those listed in Table 2, have not been successful in clinical trials.
We specifically highlight the challenges related to ALS heterogeneity, patient variability, side effects, dosage, development of biomarkers and the limitations of preclinical models. These points help provide a more balanced and comprehensive perspective on the obstacles to drug development for ALS. All modifications have been highlighted in yellow for ease of reference.
Reviewer 2 Report
Comments and Suggestions for Authors
The study titled ”Targeting protein aggregation in ALS” reviews the cover of these emerging drugs research in ALS.
Their study shows a novel review approach focusing on two strategies: addressing proteins that are prone to aggregation due to mutations and targeting the cellular mechanisms that maintain protein homeostasis to prevent aggregation. in ALS patients.
The Ms is well structured as well as their discussion suggest that the
Here, their main conclusion is that targeting protein aggregation in ALS is a promising therapeutic strategy. This is evidenced by the therapeutics currently in clinical trials that aim to prevent or reduce protein aggregation.
However, given the increase of genes associated with the disease risk and pathogenesis and in order to strengthen their discussion and conclusion, they should provide new data and explain this relevant aspect. i.e reviewed in Front Neurosci. 2019 Dec 6:13:1310.
doi: 10.3389/fnins.2019.01310
Author Response
Reviewer 2 Comments and Suggestions for Authors
The study titled ”Targeting protein aggregation in ALS” reviews the cover of these emerging drugs research in ALS.
Their study shows a novel review approach focusing on two strategies: addressing proteins that are prone to aggregation due to mutations and targeting the cellular mechanisms that maintain protein homeostasis to prevent aggregation. in ALS patients.
The Ms is well structured as well as their discussion suggest that the
Here, their main conclusion is that targeting protein aggregation in ALS is a promising therapeutic strategy. This is evidenced by the therapeutics currently in clinical trials that aim to prevent or reduce protein aggregation.
However, given the increase of genes associated with the disease risk and pathogenesis and in order to strengthen their discussion and conclusion, they should provide new data and explain this relevant aspect. i.e reviewed in Front Neurosci. 2019 Dec 6:13:1310.
doi: 10.3389/fnins.2019.01310
Authors’ response
We thank the reviewer for their insightful feedback. In response, we have modified the conclusion (lines 457-478) to strengthen our discussion regarding the several mechanisms of ALS pathogenesis, beyond proteostasis impairment. We agree that this is a critical aspect of ALS research, and we have incorporated this important point into the text. Additionally, we have added the suggested reference (now reference 78) to support the discussion. All modifications have been highlighted in yellow for ease of reference.
Round 2
Reviewer 1 Report
Comments and Suggestions for Authors
None